# LongPerceptualThoughts:
# Distilling System-2 Reasoning for System-1 Perception

**Yuan-Hong Liao**[♠]  **Sven Elflein**[♠♢]  **Liu He**[♣]  **Laura Leal-Taixé**[♢]

**Yejin Choi**[♢]  **Sanja Fidler**[♠♢]  **David Acuna**[♢]

## Abstract

Recent reasoning models through test-time scaling have demonstrated that long chain-of-thoughts can unlock substantial performance boosts in hard reasoning tasks such as math and code. However, the benefit of such long thoughts for system-2 reasoning is relatively less explored in other domains such as perceptual tasks where shallower, system-1 reasoning seems sufficient. In this paper, we introduce *LongPerceptualThoughts*, a new synthetic dataset with 30K long-thought traces for perceptual tasks. The key challenges in synthesizing elaborate reasoning thoughts for perceptual tasks are that off-the-shelf models are not yet equipped with such thinking behavior and that it is not straightforward to build a reliable process verifier for perceptual tasks. Thus, we propose a novel three-stage data synthesis framework that first synthesizes verifiable multiple-choice questions from dense image descriptions, then extracts simple CoTs from VLMs for those verifiable problems, and finally expands those simple thoughts to elaborate long thoughts via frontier reasoning models. In controlled experiments with a strong instruction-tuned 7B model, we demonstrate notable improvements over existing visual reasoning data-generation methods. Our model, trained on the generated dataset, achieves an average +3.4 points improvement over 5 vision-centric benchmarks, including +11.8 points on V* Bench. Notably, despite being tuned for vision tasks, it also improves performance on the text reasoning benchmark, MMLU-Pro, by +2 points. [1]

## 1 Introduction

Reasoning models, such as OpenAI's o1 (OpenAI et al., 2024) and Deepseek's R1 (DeepSeek-AI et al., 2025), have demonstrated remarkable capabilities in solving complex reasoning problems by scaling test-time compute. Intuitively, they increase the number of tokens generated at inference-time, allowing the model to "think longer"—producing longer chain-of-thoughts (CoTs) that go beyond typical linear rationales that mimic textbook responses.

However, despite numerous attempts to match the performance of models like o1 or R1 on challenging math benchmarks (MAA, 2024; Lightman et al., 2023), less effort has been directed toward tasks beyond mathematical reasoning. Most notably, how to generate synthetic, long-form CoT reasoning traces that solve *vision-centric tasks*—suitable for both supervised fine-tuning and preference-based RL—remains largely underexplored.

Vision-centric tasks have proven challenging for vision-language models (VLMs), especially when the tasks require object counting and localization, scene understanding, and 2D/3D spatial reasoning (Liao et al., 2024; Rahmanzadehgervi et al., 2024; Campbell et al., 2024). Prior works have addressed these challenges by helping VLMs "see" better. Common approaches include modifying the input image (*e.g.*, through cropping) or incorporating intermediate representations, into the CoT (Wu & Xie, 2024; Shao et al., 2024a; Wu et al.,

---

[1]Project website: `https://andrewliao11.github.io/LongPerceptualThoughts`
[♠]University of Toronto, Vector Institute, [♢]NVIDIA, [♣]Purdue University

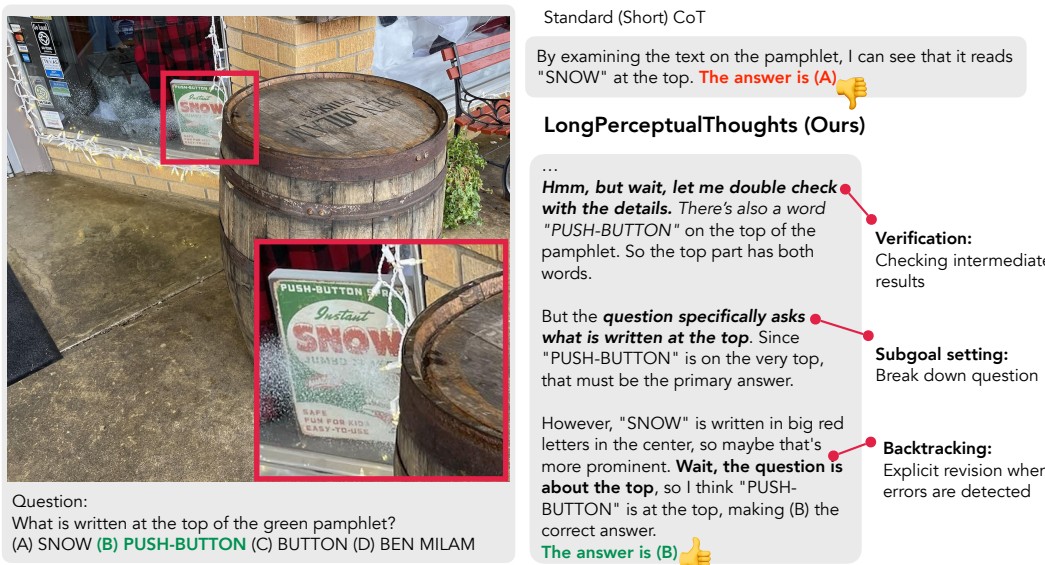

Figure 1: **LongPerceptualThoughts is a new synthetic dataset with 30K long-thought traces for vision-centric tasks.** Each trace contains diverse cognitive behaviors (e.g., verification, subgoal setting, and backtracking), akin to system-2 reasoning. CoTs generated by open-source VLMs often produce linear, rigid reasoning traces (top). In contrast, our novel data synthesis framework effectively expands these simple thoughts using frontier reasoning models, equipping VLMs with complex reasoning structures and rich cognitive behaviors—effectively distilling system-2 reasoning into instruction-tuned VLMs.

2025b). In contrast, we propose to synthesize data that implicitly equips VLMs with an internal search mechanism—one that unfolds through a textual inner monologue, enabling the model to explore multiple potential solution paths: revisiting different image regions, verifying intermediate conclusions, identifying inconsistencies, and self-correcting when necessary. Our approach is complementary to prior methods and mirrors the behavior observed in reasoning models like R1 and commercial VLMs such as o1, where reasoning performance improves by scaling test-time inference. We emphasize we do not claim that long textual CoTs are inherently superior or the only way to scale test-time inference in VLMs. Rather, our goal is to synthesize data to equip models with such capability—an approach shown to be effective in SoTA reasoning models. Furthermore, given the difficulty of building reliable process verifiers for search in perceptual tasks, our data-centric method offers a practical alternative.

In this work, we take a first step toward a scalable method that synthesizes long CoT data for vision-centric tasks. Specifically, we propose a novel three-stage data synthesis framework that: (1) generates *synthetic verifiable* multiple-choice questions from dense image captions, (2) extracts simple CoTs from VLMs for those questions, and (3) expands these simple CoTs into richer, long-form reasoning traces using frontier reasoning models. Notably, our framework performs *three layers of synthesis:* one to generate questions , one to think, and the last one to think harder. As shown in Fig. 1, using our framework, we generate **LongPerceptualThoughts**, a dataset of 30k examples for both supervised fine-tuning (SFT) and direct preference optimization (DPO), and use it to fine-tune a strong instruction-tuned VLM. The resulting model shows an average +3.4 points improvement across 5 vision-centric benchmarks, including a gain of +11.8 points on V* Bench, while typical multimodal reasoning datasets fail to improve the base VLM due to overthinking. Notably, despite being tuned for vision tasks, it also improves on the challenging text reasoning benchmark MMLU-Pro by +2 points.

## 2 Synthesize Long CoT Data for Vision-Centric Tasks

In this section, we introduce a novel data synthesis framework to synthesize long chain-of-thought (CoT) data for fine-tuning a vision-language model (VLM). Inspired by DeepSeek's R1, we are interested in collecting data consisting of thoughts and answers in the format of `<think> thought </think> <answer> answer </answer>`. We start by discussing two desired properties of reasoning data for vision-centric tasks in Sec. 2.1. Based on these two properties, we explain our data synthesis framework in Sec. 2.3. Finally, we use the synthesized long CoT data to construct LongPerceptualThoughts, consisting of both SFT and preference datasets in Sec. 2.4.

### 2.1 Desired Properties in Long Chain-of-Thought

Inspired by the recent success in OpenAI's o1 and DeepSeeks's R1, we further define *Long CoT* as an extended, structured rationale that mirrors how a human might approach complex visual reasoning tasks. Unlike the short, linear responses typically produced by current open-source VLMs, Long CoTs explore alternative solutions, verifying intermediate steps, and adjusting course when necessary. Drawing on the framework proposed in Gandhi et al. (2025), we characterize Long CoTs in vision-centric tasks through three core cognitive behaviors: **verification** (checking intermediate conclusions for correctness), **backtracking** (recognizing and revising failed solution paths), and **subgoal setting** (breaking down the task into smaller, solvable components). These cognitive behaviors have been observed in LLM to increase performance by scaling test-time compute (Muennighoff et al., 2025).

To study cognitive behaviors in vision-centric tasks, we begin by analyzing the outputs of strong instruction-tuned VLMs, following Gandhi et al. (2025). Despite its general capabilities, the model rarely displays the cognitive behaviors described earlier. The responses are often shallow and rigid, lacking the iterative, self-corrective reasoning we aim to capture. Figure 3a quantifies this gap between the response from Qwen2.5-VL-7B-Instruct and Gemini 2.0 Flash Thinking. At the end of this section, we introduce LongPerceptualThoughts that drastically diversifies the standard CoT with the desired cognitive behaviors.

### 2.2 Preliminaries

Formally, given an **image** $v$, our goal is to construct a quadruple $(v, q, Z, a)$, consisting of a **question** $q$, a long CoT **reasoning trace** $Z$, and a **final answer** $a$. We also assume the access to dense image **descriptions** $c$. A long CoT is composed of multiple thoughts that incorporate cognitive behaviors such as backtracking, verification, and subgoal setting. Formally, we define a long CoT as a sequence of intermediate thoughts: $Z := z_1 \oplus z_2 \oplus \ldots$, where $\oplus$ denotes concatenation and $z$ is a sequence of sentences, typically delineated by double new lines, *i.e.* "\n\n".

For preference data—used in reinforcement learning—our goal is to construct a preference pair of $(v, q, Z^+, a^+) \succ (v, q, Z^-, a^-)$, where the superscripts $+$ and $-$ indicate the preferred (correct) and non-preferred (incorrect or suboptimal) reasoning trajectories and their answers, and $\succ$ denotes that the left-hand tuple is preferred over the right-hand one.

### 2.3 Thought-Expansion: Distilling System-2 Reasoning into Instruction-Tuned VLMs

For an image $v$, we begin by assuming access to its dense image description $c$ that provides comprehensive visual features in the image. While in principle, one could also obtain such descriptions using a captioning model, here we assume access to such a dataset *e.g.*, DOCCI (Onoe et al., 2024) or DCI (Urbanek et al., 2023). In our proposed data synthesis framework, we leverage three foundation models: an LLM , a VLM that takes interleaved image and text as input and generates text, and a reasoning LLM that explicitly produces thoughts and answers. We use $\mathcal{M}_{\text{LLM}}$, $\mathcal{M}_{\text{VLM}}$, and $\mathcal{M}_{\text{Reason}}$ to denote them, respectively.

Below, we describe the three key stages of our data synthesis process.

**Stage 1: Convert dense descriptions to multiple-choice questions** We first convert dense descriptions into multiple-choice questions (MCQs) using an LLM. Specifically, we prompt

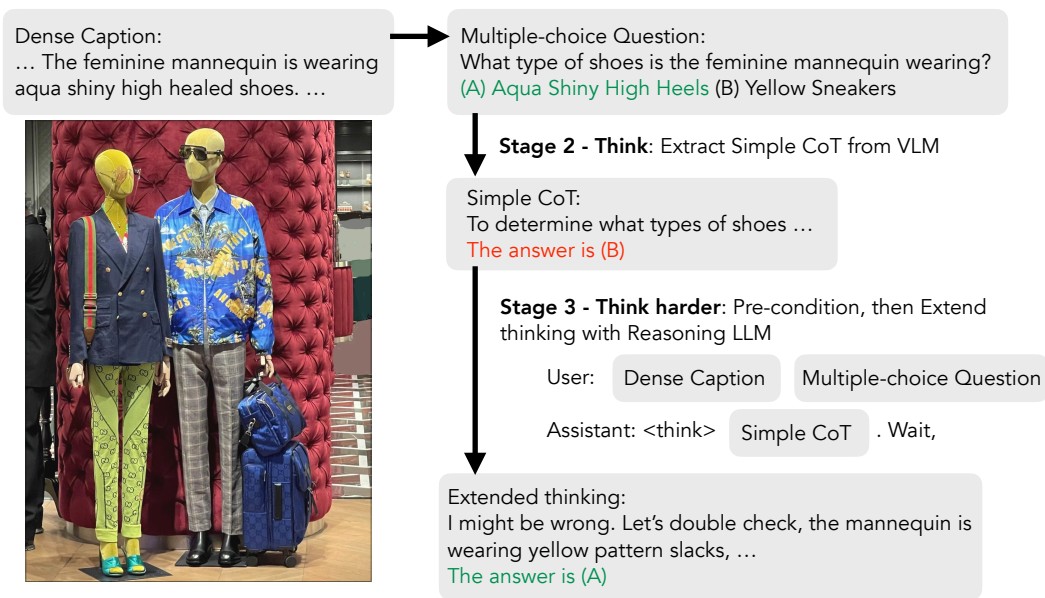

Figure 2: **Ask, Think, and Think Harder: The three stages to synthesize long CoT data for vision-centric tasks.** Assuming the access to an image and its associated dense caption, we first ask an LLM to convert dense captions to multiple-choice questions. In Stage 2, we extract simple CoT from VLM. These simple CoTs typically exhibits shallow and rigid reasoning, especially in vision-centric tasks. Therefore, in Stage 3, we precondition a reasoning LLM with these simple CoTs and append a subtle cue, *e.g.*, "Wait,", to elicit more diverse long CoTs.

$\mathcal{M}_{\text{LLM}}$ to generate MCQs based on an image and its associated dense descriptions. This step offers two key advantages that are leveraged in subsequent stages: (1) It ensures that each generated question is answerable using only the dense descriptions, allowing us to synthesize the reasoning process purely from the text modality. (2) The multiple-choice format enables easy identification of prediction correctness, which is essential for constructing positive and negative pairs in our preference dataset. Formally, this step produces a triplet $(v, q, a^\star) := \mathcal{M}_{\text{LLM}}(v, c)$. We use `gpt-4o-mini` as $\mathcal{M}_{\text{LLM}}$ to balance the cost and the quality of MCQs.

**Stage 2: Extract Simple CoTs from VLM** To generate long CoTs that the VLM is familiar with, we use the same VLM that will later be fine-tuned. Specifically, we prompt $\mathcal{M}_{\text{VLM}}$ with the image and the multiple-choice question generated in Stage 1 to produce a rationale and a final prediction, denoted as $(z_1, a_1) := \mathcal{M}_{\text{VLM}}(v, q)$. Sampling from the same VLM ensures that the synthesized CoTs remain within the model's output distribution, which we observed to be a key factor in downstream performance. By comparing the predicted answer $a_1$ with the ground-truth answer $a^\star$ from Stage 1, we can further categorize the data into positive $(z_1^+, a_1^+)$ or negative examples $(z_1^-, a_1^-)$. These can then be reused to construct either a SFT or a preference dataset. This process is akin to the rejection sampling in self-training algorithms such as RFT (Yuan et al., 2023) and STaR (Zelikman et al., 2022). We choose Qwen2.5-VL-7B-Instruct as our $\mathcal{M}_{\text{VLM}}$, as the Qwen2.5 series has demonstrated a non-trivial probability of exhibiting cognitive behaviors (Gandhi et al., 2025).

**Stage 3: Thought-Expansion using the Reasoning Model.** The analysis in Fig. 3a reveals that CoTs sampled from open-source VLMs typically exhibit shallow and rigid reasoning, with limited exploration in the output space. Inspired by the diverse cognitive behaviors observed in the responses of frontier reasoning models, we aim to leverage a reasoning LLM to generate long CoTs. However, naively sampling from $\mathcal{M}_{\text{Reason}}$ can produce CoTs that deviate significantly from the output distribution in VLM, which may degrade the performance of instruction-tuned models during fine-tuning. The similar findings have

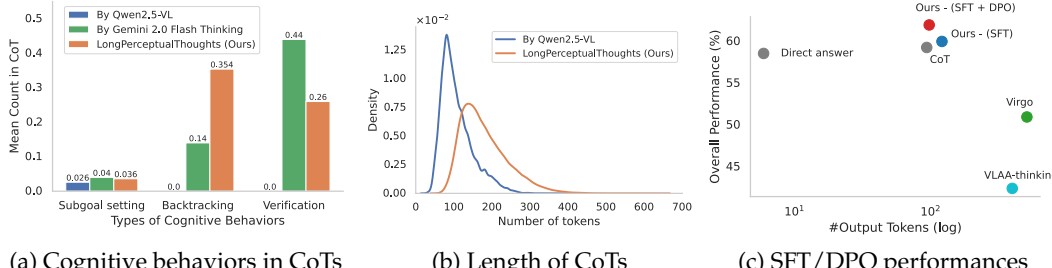

(a) Cognitive behaviors in CoTs      (b) Length of CoTs      (c) SFT/DPO performances

Figure 3: **(a) Analysis of Cognitive Behaviors in Chain-of-Thought (CoT).** CoTs from open-source VLMs often follow rigid structures. In contrast, frontier reasoning VLMs—such as Gemini 2.0 Flash Thinking—exhibit more diverse cognitive behaviors, including subgoal setting, backtracking, and verification. Our introduced long CoT dataset, LongPerceptualThoughts, also demonstrates a wide range of such behaviors. **(b) Length of CoTs.** The CoTs in LongPerceptualThoughts are significantly longer than those generated by popular VLMs, *e.g.* Qwen2.5-VL. **(c) Response length vs. aggregated performances.** Fine-tuning VLM on LongPerceptualThoughts with complex reasoning structures lead to higher overall performances with slightly more output tokens. On the other hand, fine-tuning on other multimodal reasoning leads to over-thinking and worse performance. Cognitive behaviors are quantified following Gandhi et al. (2025).

been discovered in LLMs as well (Ren et al., 2024; Li et al., 2025; Wu et al., 2025a). To address this, we introduce a *thought-expansion* mechanism that guides the reasoning LLM $\mathcal{M}_{\text{Reason}}$ to extend the CoT produced in Stage 2, while injecting cognitive behaviors such as backtracking, verification, and subgoal setting. Specifically, we precondition $\mathcal{M}_{\text{Reason}}$ with the CoT generated by VLM $z_1$ and append a subtle cue—selected from a set of predefined markers m (*e.g.*, "Wait," "Hmm," "Alternatively,")—to elicit more reflective or exploratory responses. Formally, we structure the prompt as:

$$\text{User: } c \oplus q$$
$$\text{Assistant: } \texttt{<think>} \oplus z_1 \oplus m$$

and ask $\mathcal{M}_{\text{Reason}}$ to continue the thought to obtain $(z_2, a_2)$. This approach enables the reasoning LLM to expand the familiar reasoning traces while enriching them with non-linear problem-solving behaviors. Similar to Stage 2, we can also use $a_2$ to categorize the data into positive or negative examples. Fig. 2 demonstrates the way to construct such prompt visually. We use DeepSeek-R1-Distill-Qwen-32B as our $\mathcal{M}_{\text{Reason}}$, as it is derived from the same Qwen2.5 series as the $\mathcal{M}_{\text{VLM}}$. For more details, see the prompt template in Appendix F.7.

Our proposed framework is scalable and only assumes the access to dense image description datasets. From Stage 1 to Stage 3, for an image v and its associated descriptions c, we obtain MCQs $(q, a^\star)$ in Stage 1. Then, in Stage 2 and 3, we obtain two intermediate thoughts and their associated predicted answers $(z_1, z_2, a_1, a_2)$. Finally, we have long CoT data $Z$ obtained by: $z_1 \oplus m \oplus z_2$. We will omit m in the following sections for the sake of brevity and clarity.

## 2.4 Construct SFT and DPO Datasets

In Sec. 2.3, we described a process to obtain long-form CoTs that not only aligns with the VLM's output distribution but also contains system-2 reasoning behaviors. To construct a supervised fine-tuning (SFT) dataset, we collect CoTs that lead to correct predictions. This includes examples of the form:

$$(z_1^+, a_1^+), \ (z_1^+ \oplus z_2^+, a_2^+), \ (z_1^- \oplus z_2^+, a_2^+)$$

To construct a preference dataset, we follow Setlur et al. (2024); Zhang et al. (2025) and define pairwise preferences based on correctness and compactness Team et al. (2025). Specifically:

**Correctness:**

$$(z_1^+, a_1^+) \succ (z_1^-, a_1^-)$$
$$(z_1^- \oplus z_2^+, a_2^+) \succ (z_1^-, a_1^-)$$

**Compactness:**

$$(z_1^+, a_1^+) \succ (z_1^+ \oplus z_2^+, a_2^+)$$

Akin to Setlur et al. (2024), by constructing the preference pairs of $(z_1^- \oplus z_2^+, a_2^+) \succ (z_1^-, a_1^-)$, we encourage the model to increase the likelihood $P(z_2^+, a_2^+ | z_1^-)$ and decrease the likelihood $P(a_1^- | z_1^-)$, leading to better credit assignment.

**Filtering.** Since $z_2$ is generated by a reasoning LLM using dense captions as input, it may include phrases like "As the description says." To address this, we define a list of "bad words" and filter out any thoughts containing them.

**Details of LongPerceptualThoughts.** We use 500 images and their dense captions from DOCCI. Stage 1 produces 4590 multiple-choice questions (MCQs). For long CoT data, we construct an SFT dataset with 30295 examples and a preference dataset with 17208 pairs, following filtering and deduplication. We use `gpt-4o-mini` as $\mathcal{M}_{\text{LLM}}$, Qwen2.5-VL-7B-Instruct as $\mathcal{M}_{\text{VLM}}$, and R1-Distill-Qwen-32B as $\mathcal{M}_{\text{Reason}}$.

## 3 Experiments

In this section, we first describe the experimental setup on five vision-centric benchmarks in Sec. 3.1 and present our main results in Sec. 3.2. In Sec. 3.3, we go beyond vision by evaluating our fine-tuned VLMs on a challenging text-only benchmark. Lastly, in Sec. 3.4, we analyze the response of the fine-tuned VLMs.

### 3.1 Setup

**Model.** We use Qwen2.5-VL-7B-Instruct (Bai et al., 2025) as our base model to fine-tune throughout the paper. For the sake of brevity, we refer to it as `BaseModel` in this section. We adopt full-parameter fine-tuning using LLaMA-factory (Zheng et al., 2024). See more training details in Appendix F.

**Benchmarks.** We evaluate our models on vision-centric tasks. For benchmarks covering general knowledge, we only keep their vision-centric splits, such as MME-RealWorld (Zhang et al., 2024) and MMStar (Onoe et al., 2024). To better clarify the differences, we refer to them as MME-RW-V and MMStar-V, respectively. Additionally, following Tong et al. (2024a), we include the vision-centric benchmarks: CV-bench, V* Bench, and MMVP, that involve 2D/3D spatial reasoning, fine-trained attribution, coarse scene understanding, *etc*. In total, the benchmarks consist of 10284 images and 15315 questions. More details are in Appendix B.

**Evaluation metrics.** All the benchmarks used in this work are in multiple-choice question format. We standardize their format and use regex to extract the answers. We report accuracy across all benchmarks.

**Baselines.** To explore the vision-centric capabilities of `BaseModel`, we evaluate its zero-shot predictions and apply a prompt-based chain-of-thought approach. Specifically, we prompt the model to generate `<think> thought </think>` before producing an answer—a method we refer to as Internal Thinking CoT.

For multimodal datasets, we compare LongPerceptualThoughts with one captioning dataset, DOCCI, and two multimodal reasoning datasets, Virgo (Du et al., 2025) and VAAL-thinking (Chen et al., 2025). (1) DOCCI is a human-annotated dense caption dataset, highlighting comprehensive descriptions for images. For a fair comparison with LongPerceptualThoughts, we use the exact same set of 500 images and their captions as training data. (2) Virgo distills reasoning capabilities from the language model QwQ (Team, 2024b) and the multimodal model QvQ (Team, 2024a). We adopt Virgo's self-distillation split, generated

| Approach | Avg | CV-Bench | V* Bench | MMVP | MMStar-V | MME-RW-V |
|---|---|---|---|---|---|---|
| Qwen2.5-VL-7B-Instruct | 58.47 | 74.74 | 48.51 | 73.67 | 63.73 | 31.68 |
| + Internal Thinking CoT | 59.18 | 75.42 | 55.08 | 70.60 | 62.40 | 32.40 |
| + DOCCI | 36.14 | 50.82 | 39.96 | 48.67 | 8.67 | 32.58 |
| + VLAA-thinking | 42.32 | 68.50 | 53.53 | 66.67 | 0.53 | 22.38 |
| + Virgo | 50.87 | 67.22 | 44.14 | 57.67 | 57.60 | 27.71 |
| + LongPerceptualThoughts- SFT **(Ours)** | 59.90 | 76.05 | **60.53** (+12.02) | 70.00 | 60.67 | 32.25 |
| + LongPerceptualThoughts- SFT + DPO **(Ours)** | **61.87** (+3.4) | **76.61** (+1.8) | 60.31 (+11.8) | **75.00** (+1.33) | **64.00** (+0.27) | **33.45** (+1.77) |

Table 1: **Main results on five vision-centric benchmarks.** We group the approaches into three categories: training-free methods, existing multimodal reasoning datasets, and our proposed LongPerceptualThoughts. On vision-centric tasks, fine-tuning on other multimodal reasoning datasets often leads to poorer performance, likely due to reduced instruction-following ability, domain mismatch, or an inability to capture the complex reasoning learned by larger models. In contrast, fine-tuning on LongPerceptualThoughts yields an average improvement of +1.5 points, and this gain increases to +3.4 points when using preference pairs. Notably, it achieves a 12-point improvement on the challenging V* Bench.

by first distilling QwQ into Qwen2-VL-72B-Instruct, then using the fine-tuned model to produce reasoning data for multimodal questions. The Virgo dataset includes $14,540$ examples. (3) VLAA-thinking generates multimodal reasoning data by prompting DeepSeek's R1 model with additional caption information. It contains 158k examples, from which we randomly sample 25k for training to match a similar size to our dataset. [2]

## 3.2 Main Results

We report aggregated performances in Table 1 and detail our main findings on five vision-centric benchmarks:

**LongPerceptualThoughts consistently improves performance on vision-centric benchmarks by +3.4 points via DPO.** We first perform supervised fine-tuning on the synthesized LongPerceptualThoughts. Across 5 benchmarks, we improve `BaseModel` by nearly +1.5 points on average, especially in challenging tasks such as V* bench, improving by +12 points. However, the improvements on the rest of the benchmarks are marginal. We hypothesize that this is due to noisy or erroneous tokens in our SFT datasets, which may hurt fine-tuning performance. While several prior works suggest the impacts of such erroneous tokens are marginal, they either work on models under 300M parameters (Ye et al., 2024) or target different aspects, such as safety alignment (Zhang et al., 2025). In this work, we try not to over-engineer the training recipe to highlight the effectiveness of the synthesized datasets. Unlike VLAA-thinking and Virgo that simply distill knowledge from reasoning LLMs or VLMs, our data generation pipeline allows us to construct preference data. By fine-tuning on these preference pairs, the aforementioned erroneous tokens might naturally be mitigated. For example, by performing preference-based fine-tuning such as DPO, on $(z_1^- \oplus z_2^+, a_2^+) \succ (z_1^-, a_1^-)$, the model should naturally increase the likelihood of $P(z_2^+, a_2^+|z_1^-)$ as opposed to $P(a_1^-|z_1^-)$. This helps the model reduce the impact of erroneous tokens. We find that by first applying SFT and then DPO, we obtain consistent improvements across all evaluation datasets, improving by +3.4 accuracy points. Overall, we find that LongPerceptualThoughts generally leads to consistent improvements and the preference data is the key to bring up the improvements.

**Off-the-shelf captioning data hurts instruction-tuned VLMs on vision-centric benchmarks.** Since LongPerceptualThoughts is derived from DOCCI, we are interested to see if fine-tuning `BaseModel` on DOCCI improves. Table 1 shows that training on DOCCI results in inferior performances. Perhaps expected, we find that fine-tuning on DOCCI alone especially leads to bad instruction following.

**Off-the-shelf distillation hurts performance on vision-centric benchmarks.** Both Virgo and VLAA-thinking are multimodal reasoning datasets. VLAA-thinking is distilled from

---

[2]We accessed the dataset in mid-March 2025.

R1 with the help of image captions. Virgo is distilled from a fine-tuned VLM distilled from QwQ. While both datasets are equipped with complex reasoning structures, fine-tuning the `BaseModel` does not improve vision-centric performance; instead, finetuning on VLAA-thinking and Virgo hurts the performances by -16.15 and -7.6 points, respectively. We hypothesize that there are two reasons that lead to the performance drops: (1) Both datasets are distilled from a much much larger LLMs (671B R1 model) or VLMs (Qwen2-VL-72B-Instruct), potentially resulting in the learnability gap (Li et al., 2025). (2) In particular, the multimodal reasoning data from Virgo is math-focused. We hypothesize that there is a gap between reasoning over perceptual tasks and math-related tasks. On the other hand, VLAA-thinking consists of a diverse set of datasets including DocVQA (Mathew et al., 2021), ChartVQA (Masry et al., 2022), *etc*. When using reasoning data exclusively from more natural image sources, we surprisingly observe worse performance than random subsampling. See Appendix D for details.

### 3.3 Beyond Vision: Evaluation on the Text-Only Reasoning Benchmark

Following the same setup in Sec. 3.1, we evaluates VLMs fine-tuned on multimodal reasoning training datasets in out-of-distribution (OOD) tasks. In particular, we adopt MMLU-Pro, a challenging text-only reasoning benchmark.

**MMLU-Pro (Wang et al., 2024a).** MMLU-Pro is built on top of MMLU (Hendrycks et al., 2021) by integrating more reasoning-focused questions and expanding the choices set. MMLU-Pro spans 14 diverse domains including mathematics, physics, chemistry, *etc*., encompassing over 12000 questions.

**Results.** As shown in Table 2, we find that `BaseModel` fine-tuned on LongPerceptualThoughts surprisingly improves on these text-only reasoning tasks, with an average gain of +2 points. In contrast, VLAA-thinking and Virgo hurt performance, suggesting that directly distilling from stronger teachers may lead to sharp drops in OOD tasks. We propose two hypotheses for LongPerceptualThoughts' effectiveness: (1) it introduces complex reasoning structures

| Approach | Acc |
|---|---|
| Qwen2.5-VL-7B-Instruct | - |
| + CoT | 48.07 |
| + DOCCI | 32.99 |
| + VLAA-thinking | 21.56 |
| + Virgo | 37.95 |
| + **Ours** - SFT | **50.77** |
| + **Ours** - SFT + DPO | 50.20 |

Table 2: **Evaluation on out-of-distribution tasks text-only reasoning benchmark MMLU-Pro.**

that improve `BaseModel`'s general reasoning abilities; and (2) it remains close to the original output distribution, making the new reasoning skills easier to learn without disrupting existing knowledge. Additional MMLU-Pro evaluation details are provided in Appendix E.

### 3.4 Analysis on Fine-tuned VLM Responses

To better understand our fine-tuned VLM, we visualize its responses with two key factors: aggregated performances and question difficulties.

**Response length vs. performances.** There has been growing interest in how LLMs leverage test-time compute. To investigate this, we aggregated response lengths and performance across five vision-centric benchmarks. Fig. 3c illustrates the relationship between test-time compute—measured by response length—and model performance. We observe that VLMs fine-tuned on LongPerceptualThoughts tend to produce slightly longer responses, especially after SFT. Interestingly, DPO training results in shorter responses, which aligns with the compactness encouraged during DPO pair construction. One possible direction is to exclude such preference pairs to allow models to make fuller use of test-time compute. In contrast, other multimodal reasoning benchmarks reveal signs of overthinking, where models generate unnecessarily lengthy responses.

**Response length vs. question difficulty.** Another desirable characteristic of the thinking process in LLMs is their ability to allocate appropriate "thinking time" based on a question's difficulty. Following prior works (Lightman et al., 2024; Snell et al., 2025), we define question difficulty with respect to a base VLM, i.e., Qwen2.5-VL-7B-Instruct. For each question, we estimate the model's accuracy using 11 samples and bin the questions into two quantiles:

easy and hard. Our analysis focuses on the outputs of the VLM fine-tuned via DPO on LongPerceptualThoughts. We observe that the model naturally allocates more test-time compute—reflected in longer responses—for harder questions, where its original (pre-fine-tuning) performance was worse. See Appendix C for details and visualization.

## 4    Related Work

**Reasoning in Large Language Models.** The complex reasoning abilities of large language models (LLMs) have been uncovered through various approaches. Chain-of-thought (CoT) prompting elicits their intrinsic reasoning capabilities, improving performance on language-based causal reasoning tasks (Wei et al., 2022), and has been extended into tree-based searches to enhance reasoning further (Yao et al., 2023). Similar search-like behavior can be induced through verifier guidance (Lifshitz et al., 2025), curated datasets (Shao et al., 2024b), or supervised fine-tuning on synthetic reasoning data (Gandhi et al., 2024; Lehnert et al., 2024). More recently, DeepSeek-R1 (Guo et al., 2025) achieved state-of-the-art reasoning through reinforcement learning, exhibiting human-like traits such as self-correction and verification. Other open-source efforts have been explored to study various tricks to scale reinforcement learning Liu et al. (2025a); Pan et al. (2025). In contrast, s1 (Muennighoff et al., 2025) and LIMO  (Ye et al., 2025) improves mathematical reasoning via supervised fine-tuning on less than 1000 distilled reasoning traces. While most prior work focuses on math and coding tasks, our goal is to explore how such strong reasoning capabilities can be effectively applied to perception.

**Reasoning in Vision-Centric Tasks.** Unlike reasoning in math or coding tasks, vision-centric problems often involve significant uncertainty due to partial information, perceptual noise, and visual ambiguities. Prior works primarily address this by helping VLMs "see" better. For instance, SEAL (Wu & Xie, 2024) uses a search-like cropping mechanism to iteratively navigate an image, while VisualCoT (Shao et al., 2024a) generates auxiliary visual cues to guide attention. Other approaches (Wang et al., 2024b; Liao et al., 2025) decompose complex tasks into simpler verification steps to enhance model robustness. In contrast, we aim to teach VLMs to reason better—encouraging them to explore multiple solution paths by revisiting image regions, verifying intermediate conclusions, and engaging in textual inner monologue. Concurrent work on multimodal reasoning addresses this challenge, particularly in math problem solving, using techniques such as distillation from advanced reasoning LLMs (Du et al., 2025; Thawakar et al., 2025) and reinforcement learning (Liu et al., 2025b; Huang et al., 2025). In this work, we study how system-2 reasoning can improve vision-centric tasks, and propose a data synthesis framework that generates long CoT examples to teach visual reasoning through deliberate, step-by-step thinking in the textual space.

## 5    Conclusions

In this work, we explore how system-2 reasoning can enhance vision-centric tasks. We introduce a novel, scalable data synthesis framework that requires only dense image captions. The framework generates verifiable multiple choice questions, extracts simple chains of thought (CoTs) from vision-language models (VLMs), and expands them into rich, long-form reasoning traces using frontier reasoning models. This process yields LongPerceptualThoughts, a synthetic dataset containing 30k detailed reasoning traces for perceptual tasks. Fine-tuning Qwen2.5-VL-7B-Instruct on LongPerceptualThoughts improves performance by +3.4 points across five vision benchmarks, including an +11.8-point gain on V$^*$ Bench. Remarkably, despite being trained on vision tasks, the model also improves by +2 points on the out-of-distribution text-only reasoning benchmark MMLU-Pro.

## Acknowledgements

We thank Rafid Mahmood, Jaehun Jung, Jen-Hao Cheng, Ali Hatamizadeh, Ximing Lu, Hyunwoo Kim and Amlan Kar for their helpful comments and feedback on an early discussions and draft of this paper.

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
