# OpenReview forum: "LongPerceptualThoughts: Distilling System-2 Reasoning  for System-1 Perception"
_colmweb.org/COLM/2025/Conference — COLM 2025_

### Official Review · Reviewer_V9Qt · 2025-04-13

**Rating:** 4
**Confidence:** 2
**Ethics Flag:** 1

**Summary:**

This paper proposes to improve VLMs on difficult vision tasks through long CoTs. To do so, they synthesize a new dataset of long reasoning traces which then they use to fine-tune a Qwen model.

**Questions To Authors:**

See W1-4 above.

(I am very unlikely to improve my score because of the "main reason to reject" stated above.)

**Reasons To Accept:**

Strengths
--

(S1) Scalable automated pipeline to generate the long CoT. First using an LLM to turn captions into MC questions, then frontier reasoning models to generate long-form reasoning traces.

(S2) Improvements on a variety of tasks.

**Reasons To Reject:**

Weaknesses
--

(W1) The title doesn't make sense given my understanding of the paper (maybe I misunderstand the whole paper, in which case the text has to be improved, not the title).

- "*System-2 Reasoning for System-1 Perception*". What the authors do is to use a system-2 approach to solve vision-centric (perception) tasks that are usually addressed with a system-1 approach.

- "*Distiling*": I don't see where there is a distillation happening here. The authors use the term in the text, e.g. "Distill Simple CoT from VLM", but I don't understand this usage of the term. I am familiar with the meaning of distillation in machine learning where a teacher model's output are used as supervision to train a student model. It seems here that a better word would simply "*synthesize*"? (instead of "*distill*")

(W2) "*verifiable multiple-choice questions*": whatis verifiable? This is mentioned multiple times but it's never made clear what this means precisely, nor how this is used or enforced.

(W3) There is little insight on exactly how/why the improvements are obtained. Which key aspects of the proposed data/generation pipeline are important? There is pretty much no support to demonstrate that specific aspects of the whole section 2 (and which aspects in particular) are important.

(W4) Section 3.4 looks at response length. It notes that they are only slightly longer. I thought that the point of this paper was to enable longer reasoning? Shouldn't we expect the responses to always be **much** longer?

---

Please proofread the paper as there are quite a few typos. Here are a few ones:

L333 "*reqrequires*"

L340 "*alsothe*"

L341 "*by +2 points by +2 points*"

L236/240: repeated paragraph header? (almost the same)

L281 *performances* -> *performance* (singular)

---

Main reason to reject: no scientific analysis or take-aways. The authors describe a method, evaluate it and show some improvements. No ablations etc. Not sufficient for publication in its current form, but we encourage the authors to extend this work and this could become a very interesting paper.

---

> ### Author Response · Authors · 2025-06-02
> **Response to Reviewer V9Qt**
>
> We thank the reviewer for the thoughtful and constructive feedback. We're encouraged by your recognition of the scalable, automated pipeline we propose and the demonstrated improvements across diverse tasks. We acknowledge your concerns regarding writing and analysis; however, other reviewers have positively recognized this aspect of our work. For instance:
> - Reviewer DSU2 says that *"the paper has conducted insightful analysis of where the performance improvement comes from ..."*
> - Reviewer EHkH says that *“the writing is very clear and easy to follow, the experiments are thorough and complete ...”*
>
> Below, we address each of your concerns:
>
> 1. Clarification of the title
>
> > What the authors do is to use a system-2 approach to solve vision-centric (perception) tasks that are usually addressed with a system-1 approach.
>
> Yes, this is exactly our intention. Our goal is to tackle traditionally system-1 perception tasks using explicit, structured, system-2-style reasoning. We will revise the abstract and introduction to make this goal more transparent early on.
>
> > I don't see where there is a distillation happening here ... It seems here that a better word would simply "synthesize"?
>
> We agree that the term "distill" may be confusing. To clarify: distillation occurs in Stage 3, where a reasoning LLM (R1-Distill-Qwen-32B; see Appendix L676) injects structured reasoning into our dataset. We will revise the text to use "synthesize" for the simple CoTs from the VLM and reserve "distill" for the structured traces from R1.
>
> 2. The confusion of *verifiable* multiple-choice questions
>
> > whatis verifiable? .. it's never made clear what this means precisely, nor how this is used or enforced.
>
> By “verifiable,” we mean questions whose answers can be automatically checked via deterministic string matching against predefined options—unlike open-ended or VQA-style questions that rely on potentially noisy LLM-based judgments
>
> This property is essential to our pipeline: in Section 2.4, we use these verifiable questions to determine the correctness of ($a_1$, $a_2$, $z_1$, $z_2$), which is critical to construct SFT and preference data.
>
> 3. Key aspects of the proposed generation pipeline
>
> > Which key aspects of the proposed data/generation pipeline are important?
>
> Thank you for pointing this out. Our pipeline’s effectiveness stems from the interplay of three stages:
> - Stage 1: generating verifiable multiple-choice questions from image captions
> - Stage 2: using VLMs to produce simple CoTs aligned with their output style
> - Stage 3: refining these with a reasoning LLM to inject deeper cognitive structure.
>
> Each stage contributes uniquely, enabling both reliable evaluation and progressively richer reasoning.
>
> To mitigate the reviewer's concern, We direct the reviewer to Table 1: LongPerceptualThoughts is compared with two other visual reasoning datasets, VL-Thinking [1] and Virgo. We note that VL-Thinking distills data from R1 followed by rewriting and filtering steps, yet still yields negative SFT performance, even on in-domain tasks. In contrast, our approach demonstrates that data synthesized using our pipeline can support effective SFT, suggesting the benefits of our framework for vision-language model training.
>
> [1] Chen et al., SFT or RL? An Early Investigation into Training R1-Like Reasoning Large Vision-Language Models
>
> *We would like to clarify that the dataset referred to as VLAA-thinking in [1] is a renamed and expanded version of the VL-thinking dataset used in our submission.*
>
> 4. Analysis of response lengths in Section 3.4
>
> > ... Shouldn't we expect the responses to always be much longer?
>
> This is an excellent observation. Our paper’s goal is to enable long-form reasoning, and we address this in two parts:
>
> 1) Are the LongPerceptualThoughts responses longer than vanilla VLM outputs?
>
> Yes. We compared our dataset to Qwen2.5-VL’s generations on DOCCI, and the LongPerceptualThoughts responses are longer on average. Please refer to this [external figure](https://i.ibb.co/LzNMjpZJ/Screenshot-2025-05-31-at-2-47-35-PM.png)
>
> 2) Does fine-tuning lead to longer outputs?
>
> Interestingly, not always. As noted in L165, our DPO training balances correctness and brevity. Thus, even if the model has access to longer CoTs, it may choose more concise outputs in certain cases. Indeed, Figure 3b shows that SFT-trained models tend to generate longer responses, while DPO-trained ones remain relatively compact despite better performance.
>
> 5. Toward a more comprehensive study
>
> > Main reason to reject: ... No ablations etc. Not sufficient for publication in its current form, ...
>
> While we agree that a comprehensive ablation of hyperparameters in our synthetic data pipeline would strengthen the paper and plan to pursue this in future work, it is not feasible during the rebuttal period due to time and budget constraints.
>
> 6. Typos
>
> We will revise the paper accordingly.

---

> > ### Comment · Reviewer_V9Qt · 2025-06-03
> > **Thanks for the rebuttal**
> >
> > I appreciate the response of the authors. On the one hand, I am glad that I didn't misunderstand the contributions of the paper. On the other hand, this means that my concerns were valid. The misleading wording would need clarifications throughout the paper. But more importantly, the study still lacks an ablative study to value of each technical contributions and the claim that the "*pipeline’s effectiveness stems from the interplay of three stages*". As the authors say, it's unreasonable to expect such work to be done within the rebuttal timeline, but I think this is critical to support the whole paper. If the authors have not conducted these ablations, I don't think that they have the evidence even for themselves that the proposed pipeline is indeed working for the reasons claimed it does (verifying this is the point of such a study). I believe this extra bit of work is critical. If its results pan out, this paper could be a good submission to a future conference.

---

> > ### Author Response · Authors · 2025-06-06
> > **Response to Reviewer V9Qt**
> >
> > We thank the reviewer for the continued discussion and would like to respond to the two remaining concerns.
> >
> > In short, we provide clarifications regarding (1) the use of the word “distill” in the title, and (2) the role of ablations in our data-centric setup. We elaborate on each point below.
> >
> > ## Use of “distill” in the title
> >
> > We appreciate the reviewer’s feedback and agree that “synthesize” is a more precise description of Stage 2; we will revise the manuscript to reflect this. That said, we respectfully disagree with the claim that the title “doesn't make sense.” **The term “distill” still accurately describes the overall pipeline**—especially Stage 3 where the CoT is extended using a larger reasoning model. Moreover, we note that other reviewers did not express confusion about the title, and Reviewer EHkH explicitly praised the clarity of our writing. While we are open to improving wording in the final version, we believe that differing interpretations of broadly-used terminology like “distill” should not, on their own, constitute grounds for rejection.
> >
> > ## Ablations
> >
> > We recognize that ablation studies are important for scientific progress, particularly when developing models or algorithms. However, our work adopts a *data-centric* perspective, where the focus is on how data is synthesized and distilled. We emphasize that our proposed data synthesis pipeline is designed to **function holistically**. As a result, modifying individual components in isolation could affect  subsequent stages, making the interpretation of such ablations more complex and potentially misleading.  That said, If the reviewer has specific suggestions for an experiment that could shed light on a specific component or assumption, we would be happy to consider incorporating for the camera ready  improving the clarity and robustness of our findings.
> >
> > **Note**: We emphasize that our presentation is not unusual. Influential works such as ShareGPT4V [1] and SpatialVLM [2] introduce holistic synthetic data generation pipelines that have significantly shaped the community’s understanding of how to synthesize data for vision-language models. Yet neither of these works includes the type of ablation the reviewer is suggesting. In holistic, data-centric setups like ours, the cost and time required for such ablations may not be justified by the insights they yield—since removing individual components in isolation can disrupt the pipeline and lead to misleading conclusions.
> >
> >
> > In the followings, we highlight **our efforts in justifying the proposed data synthesis pipeline**:
> >
> > **Our efforts in analysis**: To offer insight into the effects of our pipeline design, we compare against alternative synthesis methods and present dataset-level analyses (e.g., Fig. 3a and the additional analysis [(same link here)](https://i.ibb.co/LzNMjpZJ/Screenshot-2025-05-31-at-2-47-35-PM.png) included in our first-round rebuttal). We believe these analyses provide a complementary view of the data characteristics and their downstream implications.
> >
> > **Our efforts in comparative analysis**: We would like to clarify that our dataset, LongPerceptualThoughts, has been directly compared against two existing visual reasoning datasets—VL-Thinking and Virgo—as shown in Table 1. Notably, VL-Thinking, which is derived from R1 through rewriting and filtering, still results in negative SFT performance—even on in-domain tasks. In contrast, our pipeline produces data that enables positive and effective SFT outcomes, underscoring the advantages of our synthesis framework for training vision-language models.
> >
> > [1] Chen et al., ShareGPT4V: Improving Large Multi-Modal Models with Better Captions
> >
> > [2] Chen et al., SpatialVLM: Endowing Vision-Language Models with Spatial Reasoning Capabilities

---

> > > ### Author Response · Authors · 2025-06-10
> > >
> > > We just wanted to follow up on our earlier response. We really appreciate the reviewer's engagement and thoughtful feedback. We’d be happy to continue the discussion or clarify anything further if helpful.
> > >
> > > Thanks again!

---

### Official Review · Reviewer_WtLJ · 2025-05-11

**Rating:** 4
**Confidence:** 4
**Ethics Flag:** 1

**Summary:**

The paper introduces a three-stage data synthesis framework designed to enhance visual reasoning in vision-language models (VLMs) by generating long chains of thought (CoTs) data. The framework synthesizes verifiable multiple-choice questions from dense image descriptions, extracts simple CoTs from VLMs, and expands these CoTs into elaborate reasoning traces using advanced reasoning models. The authors demonstrate significant performance improvements on several vision-centric benchmarks, including a notable gain on the V* Bench.

**Questions To Authors:**

- How does the performance of LongPerceptualThoughts compare with methods that modify input images or incorporate intermediate representations into CoTs? A detailed comparison would help to better understand the unique contributions of the LongPerceptualThoughts.
- Can the authors provide a more comprehensive ablation study to validate the effectiveness of each component of the three-stage synthesis framework? This would include varying the number of multiple-choice options in Stage 1 and testing different trigger words in Stage 3.
- What measures are taken to prevent overfitting to existing CoTs during the thought expansion process? How can the framework ensure that it explores truly diverse thought patterns rather than reinforcing existing ones?
- How does the framework ensure that the generated CoTs remain within the model's original output distribution? Could the authors provide more details on the validation process for this aspect?

**Reasons To Accept:**

- Presents a data-centric method for enhancing visual reasoning by aiming to distill complex, system-2-like reasoning long CoTs into the synthesis dataset.
- Demonstrates the three-stage synthesis framework's effectiveness through substantial performance improvements on multiple benchmarks.
- The proposed dataset synthetic method could improve the model’s performance on challenging text reasoning benchmarks, offering a promising direction for future research.

**Reasons To Reject:**

- Lacks comparisons to existing visual reasoning methods related to CoT, like those modifying inputs or using intermediate CoT representations, which makes it difficult to identify the novelty/advantages/overall positioning of the proposed method within the existing literature.
- The ablation study presented is limited, failing to clearly delineate the individual contributions of each component within the proposed framework.
- The key process of thought-expansion is not well justified. When the generated CoTs distributions remain within the simple CoTs distribution from VLM, how could it diversify the cognitive behaviors?

---

> ### Author Response · Authors · 2025-06-02
> **Response to Reviewer WtLJ (1/2)**
>
> We appreciate the reviewer’s positive feedback on our data-centric approach, the significant performance gains across multiple benchmarks, and the improvements on the challenging text reasoning benchmark.
>
> In the followings, we address the reviewer's concerns and questions:
>
> 1. Comparison with existing CoT approach that modify inputs
>
> We appreciate the reviewer’s comment and clarify that our work focuses on implicitly equipping VLMs with an internal search mechanism (L37–L41), differing from prior methods [1,2] that modify input images to enhance perception. These approaches are conceptually distinct and complementary to ours.
>
> To directly address the reviewer’s point, we evaluated VisCoT [1] on V* Bench using the official codebase ([GitHub link](https://github.com/deepcs233/Visual-CoT)). For a fair comparison, we selected VisCoT-7b-336, which is similar in size to Qwen2.5-VL-7b-Instruct, in particular [VisCoT-7b-336](https://huggingface.co/deepcs233/VisCoT-7b-336).
>
> As shown in the table below, VisCoT-7b-336 slightly outperforms Qwen2.5-VL-7b-Instruct. However, after applying SFT followed by DPO using our proposed LongPerceptualThoughts dataset, we observe a +7 point improvement over VisCoT.
>
> We hope this result helps clarify that while our method shows promising gains, it is complementary in spirit to works like VisCoT, which take a more explicit input-modification approach (also noted in L44). We believe combining these directions may offer further potential.
>
> |                                              | V* Bench |
> |----------------------------------------------|----------|
> | Visual-CoT-7b-336                            | 53.06    |
> | Qwen2.5-VL-7b-Instruct -- Zero-shot          | 48.51    |
> | LongPerceptualThoughts - SFT-then-DPO (Ours) | 60.31    |
>
> [1] Shao et al., Visual CoT: Advancing Multi-Modal Language Models with a Comprehensive Dataset and Benchmark for Chain-of-Thought Reasoning
>
> [2] Zhang et al., MLLMs know where to look: Training-free perception of small
> details with multimodal LLMs.
>
> 2. Ablation study
>
> > The ablation study presented is limited, ... Can the authors provide a more comprehensive ablation study ... This would include varying the number of multiple-choice options in Stage 1 and testing different trigger words in Stage 3.
>
> This paper aims to enhance system-2 thinking in VLMs for perceptual tasks by synthesizing a dataset tailored for standard training methods like SFT or DPO. Inspired by the prior works (L139–140), we propose a novel three-stage data synthesis pipeline and demonstrate strong in-domain and out-of-domain gains. Design choices such as the number of options in Stage 1 and trigger words in Stage 3 are intended as tunable hyperparameters. A full ablation of these factors is beyond our current computational resources.
>
> To mitigate the reviewer's concern, we highlight the results in Table 1: LongPerceptualThoughts is compared with two other visual reasoning datasets, VL-Thinking [3] and Virgo. We note that VL-Thinking distills data from R1 followed by rewriting and filtering steps, yet still yields negative SFT performance, even on in-domain tasks. In contrast, our approach demonstrates that data synthesized using our pipeline can support effective SFT, suggesting the benefits of our framework for vision-language model training.
>
> [3] Chen et al., SFT or RL? An Early Investigation into Training R1-Like Reasoning Large Vision-Language Models
>
> *We would like to clarify that the dataset referred to as VLAA-thinking in [3] is a renamed and expanded version of the VL-thinking dataset used in our submission.*

---

> > ### Author Response · Authors · 2025-06-02
> > **Response to Reviewer WtLJ (2/2)**
> >
> > 3. Justification of thought-expansion
> >
> > >  When the generated CoTs distributions remain within the simple CoTs distribution from VLM, how could it diversify the cognitive behaviors?
> >
> > > What measures are taken to prevent overfitting to existing CoTs during the thought expansion process? How can the framework ensure that it explores truly diverse thought patterns rather than reinforcing existing ones?
> >
> > Thank you for your insightful question. If we understand correctly, your concern is whether the thought expansion process meaningfully diversifies cognitive behaviors, or merely reinforces patterns already present in the simple CoTs (Stage 2). Please let us know if we have misunderstood.
> >
> > To address this, we compare the expanded traces in LongPerceptualThoughts with the original simple CoTs along two axes:
> >
> > - **Diversity of cognitive behaviors**: As shown in Fig. 3a, Qwen2.5 VL’s simple CoTs (Section 2.3) rarely exhibit cognitive behaviors on perceptual tasks. In contrast, LongPerceptualThoughts demonstrates a richer behavioral range, comparable to proprietary VLMs like Gemini 2.0 Flash Thinking—highlighting the effectiveness of our expansion process in injecting meaningful behavioral diversity.
> > - **Token length distribution**: LongPerceptualThoughts traces are on average 100 tokens longer, with many exceeding 300 tokens, indicating deeper and more elaborate reasoning compared to the original concise CoTs. Please refer to the [external figure](https://i.ibb.co/LzNMjpZJ/Screenshot-2025-05-31-at-2-47-35-PM.png)
> >
> > 4.  Do Generated CoTs remain within the model's original output distributions?
> >
> > > How does the framework ensure that the generated CoTs remain within the model's original output distribution? ...
> >
> > The main goal of this paper is to **enhance system-2 thinking in VLMs when facing perceptual tasks**. To achieve this using standard training recipes (e.g., SFT or DPO), we study how to synthesize a dataset that facilitates this goal. Building on insights from prior works (as mentioned in L139–140), we design a novel three-stage data synthesis pipeline and demonstrate notable in-domain and out-of-domain gains.
> >
> > While it would be valuable to study whether the synthesized traces lie within the VLM's original output distribution—and to analyze how that relates to performance gains—this is out of the scope of our work.
> >
> > To reiterate, our objective is to inject system-2 thinking for perceptual tasks, rather than to ensure that the synthesized data strictly adheres to the VLM’s original output distribution.

---

> > > ### Comment · Reviewer_WtLJ · 2025-06-06
> > >
> > > Thank you to the authors for the response. The revision partially addresses Q1 and Q3, but key issues remain.
> > >
> > > For Q1, the method is only compared to one baseline (VisCoT) on a single benchmark, which limits the strength of the evidence.
> > >
> > > For Q3, while the results are promising, it remains unclear how the method enhances diverse CoTs. Most importantly, the lack of ablation studies and output analysis makes it difficult to identify what drives the improvements. Without these, the contribution is unclear and offers limited value to the research community.
> > >
> > > I recommend adding more empirical evidence—especially ablations (Q2) and deeper analysis (Q4)—to better support the claimed contributions. For now, I will maintain my score.

---

> ### Author Response · Authors · 2025-06-09
> **Response to Reviewer WtLJ -- Q1 and Q3**
>
> We thank the reviewer for the thoughtful feedback. Below, we address the concerns of Q1 and Q3:
>
> ### Q1 (Additional Baselines and Benchmarks)
>
> We have now conducted additional experiments on **CV Bench** and **MMVP**. As shown in the table below, our method consistently outperforms baselines across these diverse benchmarks, further supporting the generalizability of our approach. If the reviewer has another method in mind in addition to Visual-CoT, please let us know and we will add it.
>
>
> |                                              | V* Bench |CV Bench  | MMVP     |
> |----------------------------------------------|----------|----------|----------|
> | Visual-CoT-7b-336                            | 53.06    |63.52 | 63 |
> | Qwen2.5-VL-7b-Instruct -- Zero-shot          | 48.51    |74.74 | 73.67 |
> | LongPerceptualThoughts - SFT-then-DPO (Ours) | 60.31    |76.61 | 75 |
>
> ### Q3 (Diverse Chain-of-Thoughts)
>
> We thank the reviewer for acknowledging the promising results. To better emphasize/visualize significant increase in token length in the [external figure](https://i.ibb.co/LzNMjpZJ/Screenshot-2025-05-31-at-2-47-35-PM.png), we dump their statistics in table format:
>
> | Token Length Statistic | LongPerceptualThoughts (Ours)   | Qwen2.5-VL      |
> |-----------|---------------|---------------|
> | mean      | 179.78    | 108.43    |
> | 25%       | 134    | 78     |
> | 50%       | 167    | 98     |
> | 75%       | 214    | 129    |
>
>
> Regarding "how the method enhances diverse CoTs", we respectfully point out that our paper includes **two analyses that quantify the changes introduced by thought expansion**: (1) Diversity of cognitive behaviors (Fig. 3a) and (2) Token Length increase in the above table.
>
>
> If the reviewer could suggest specific metrics they believe are missing, we would be happy to incorporate such analysis.

---

> > ### Author Response · Authors · 2025-06-10
> > **Response to Reviewer WtLJ -- Q2 and Q4**
> >
> > > I recommend adding more empirical evidence—especially ablations (Q2) and deeper analysis (Q4)—to better support the claimed contributions. For now, I will maintain my score.
> >
> > To address the reviewer's concern, we provide a deeper analysis on:
> > 1) token count changes, and
> > 2) the average count of cognitive behaviors
> > before and after thought expansion, focusing on different trigger words (used in L145).
> >
> > ### Token Count Changes
> >
> > Building on the [external figure](https://i.ibb.co/LzNMjpZJ/Screenshot-2025-05-31-at-2-47-35-PM.png) provided in the first-round rebuttal, we further analyze token counts when using different trigger words. We find that trigger words introducing a pause (e.g., "Hmm," "Wait,") lead to longer CoTs. In comparison, words like "Alternatively," result in slightly shorter CoTs—but still longer than those produced by Qwen2.5-VL.
> >
> > | Model                 | Trigger Word     | Mean Token Count |
> > |----------------------|------------------|------------------|
> > | Qwen2.5-VL           | N/A              | 108.43           |
> > | LongPerceptualThoughts | Hmm,             | 187.97           |
> > | LongPerceptualThoughts | Wait,            | 192.57           |
> > | LongPerceptualThoughts | Alternatively,   | 166.16           |
> >
> > ### Average Count of Cognitive Behaviors
> >
> > Building on Fig. 3a, we further analyze the average counts of cognitive behaviors in CoTs using different trigger words. Trigger words introducing a pause (e.g., "Hmm," "Wait,") lead to more backtracking and verification. In contrast, words like "Alternatively," result in fewer cognitive behaviors overall—but interestingly, produce a relatively high frequency of subgoal setting.
> >
> > |                        | Trigger Word    | Backtracking | Subgoal Setting | Verification |
> > |------------------------|----------------|--------------|-----------------|--------------|
> > | Qwen2.5-VL             | N/A            | 0            | 0.026           | 0            |
> > | LongPerceptualThoughts | Hmm,           | 0.598        | 0.017           | 0.400        |
> > | LongPerceptualThoughts | Wait,          | 0.559        | 0.000           | 0.477        |
> > | LongPerceptualThoughts | Alternatively, | 0.071        | 0.067           | 0.049        |

---

### Official Review · Reviewer_DSu2 · 2025-05-13

**Rating:** 6
**Confidence:** 4
**Ethics Flag:** 1

**Summary:**

The paper presents LongPerceptualThoughts, a synthetic dataset of 30,000 long-form CoT traces designed to equip vision-language models (VLMs) with System-2 reasoning skills like verification, subgoal setting, and backtracking. Using a novel three-stage pipeline—generating verifiable MCQs from image captions, distilling simple CoTs, and expanding them with a frontier reasoning model—the authors fine-tune Qwen2.5-VL-7B-Instruct. This results in notable performance gains across five vision-centric benchmarks and surprising improvements on a text-only benchmark (MMLU-Pro), suggesting strong generalization.

**Reasons To Accept:**

* Compared to previous methods, the paper has successfully constructed a solid SFT dataset, where previous methods failed to correctly construct the reasoning SFT data for new domains. The performance improvement has shown improvement after SFT training.

* The data generation pipeline can be helpful for solving other similar tasks.

* The paper has conducted insightful analysis of where the performance improvement comes from and analyzed reasoning patterns such as subgoal setting, verification, and backtracking.

**Reasons To Reject:**

* Missing details and prompts used for constructing the dataset. As prompt engineering is important here for successfully constructing the dataset, it is good to put the prompts into the paper appendix. And it also becomes more clear how "Wait", "Hmm" are added, and how it affects the generation.

* The paper has only evaluated on multiple-choice questions. It would be better to evaluate other tasks other than perceptual. It would be interesting to see if the method can generalize to other reasoning tasks, or the method is only limited to the selected tasks in the paper.

---

> ### Author Response · Authors · 2025-06-02
> **Response to Reviewer DSu2**
>
> We appreciate the reviewer’s positive feedback on our dataset, the potential usefulness of our data generation pipeline for other domains, and the insightful analysis of the dataset’s reasoning patterns.
>
> In the followings, we address the reviewer's concerns and questions:
>
> 1. Details and prompts of constructing datasets.
>
> We agree that openness is important to the community. Section E.1 outlines the sampling parameters for each stage, and the prompts used are shown in Figures 5, 6, and 9. We thank the reviewer for noting the missing prompt in Stage 3 of the proposed data synthesis pipeline. We will  also be making the code available for reproducibility.
>
> Below is the prompt used in Stage 3, which we will add to the appendix.
>
> ```jinja
> User: You are a large language model that answers visual questions by generating a vivid mental image from a text description. Given a visual question along with an image description, create a detailed internal visualization of the image. Then, use this mental image to spatially reason through and answer the question.
>
> - After building the mental image from the text description, you should not explicitly referencing the text description in your internal reasoning. e.g., Avoid saying "The description states ..." within <think>...</think> block.
> - Ensure your reasoning is logically sound and leads coherently to the final answer. The steps you follow should clearly support the conclusion you reach.
> - Please provide your answer as (X), where X is the letter of the correct option.
> - Enclose your final answer within <answer> and </answer> tags.
>
> Assistant: <think> {{ simple_CoT }} {{ cognitive_phrase }}
> ```
>
> Variable explained:
> - `simple_CoT` is the extracted thought from Stage 3.
> - `cognitive_phrase` is the pre-defined cognitive cues, such as "Wait,", "Hmm,", and "Alternatively,"
>
>
> 2. Evaluation on different tasks.
>
> To address the reviewer’s request for evaluation beyond multiple-choice, perceptual tasks, we include preliminary results on DocVQA-val [1], a benchmark that better aligns with the desired criteria (non-multiple-choice, reasoning-focused). We sample the first 1,000 examples from the validation set (total: 5,349 QA pairs) and evaluate model outputs using `gpt-4o-2024-08-06` as an LLM-as-judge.
>
> As shown in the table below, fine-tuning on LongPerceptualThoughts yields a >30-point improvement in accuracy over zero-shot predictions. This result highlights the generalization benefits of our dataset to broader reasoning tasks beyond perception.
>
>
> |                                              | DocVQA-val|
> |----------------------------------------------|-----------|
> | Qwen2.5-VL -- Zero-shot                      | 23   |
> | LongPerceptualThoughts - SFT-then-DPO (Ours) | 58.4    |
>
>
> *To ensure transparency, we include the exact prompt we used for LLM-as-a-judge evaluation below:*
>
> ```jinja
> System: You are a rigorous evaluator judging whether AI model predictions match ground truth answers.
> Always respond with 'yes' or 'no' only.
>
> User: You are a strict judge evaluating an AI model's prediction.
> All Possible Ground Truth Answer: {{ ground_truth_list }}
> Model Prediction: "{{ prediction }}"
>
> Does the prediction correctly and completely match the ground truth?
> Respond only with 'yes' or 'no'.
> ```
>
> We will include evaluation results using a standardized prompt format in the camera-ready version upon acceptance.
>
>
>
> [1] Mathew et al., DocVQA: A Dataset for VQA on Document Images

---

> > ### Comment · Reviewer_DSu2 · 2025-06-05
> >
> > Thanks for the response. I don't have further questions.

---

### Official Review · Reviewer_EHkH · 2025-05-17

**Rating:** 7
**Confidence:** 5
**Ethics Flag:** 1

**Summary:**

This paper introduce LongPerceptualThoughts, a syhthetic visual reasoning dataset consisting of 30K long reasoning traces on visual perception tasks.

LongPerceptualThoughts is built based on a carefully designed 3-stage pipeline: QA generation based on dense caption with LLMs, VQA reasoning generation with vision language models and longer reasoning traces generation with text reasoning models.

Experiment results show that LongPerceptualThoughts can enhance off the shelf vision language models' perforance on vision-centric tasks by 3.4 percent. And also surprisingly improves their performance on text reasoning benchmarks (MMLU-pro) by 2 points.

**Questions To Authors:**

LongCoT can also be harmful for MLLMs since they tend to hallucinate more when its output is longer[1]. It would be great to discuss about this aspect since the dataset is endowing such ability to MLLMs.

Another relevant visual cropping/searching paper[2] you might want to discuss in related works, where they follow model's own inference time dynamics to help the model zoom into the area of interest.

[1] Huang, Q., Dong, X., Zhang, P., Wang, B., He, C., Wang, J., ... & Yu, N. (2024). Opera: Alleviating hallucination in multi-modal large language models via over-trust penalty and retrospection-allocation. In Proceedings of the IEEE/CVF Conference on Computer Vision and Pattern Recognition (pp. 13418-13427).

[2] Zhang, J., Khayatkhoei, M., Chhikara, P., & Ilievski, F. (2025). MLLMs know where to look: Training-free perception of small visual details with multimodal LLMs. arXiv preprint arXiv:2502.17422.

**Reasons To Accept:**

The reasearch question is very important. Although straightforward, fine-grained visual perception is a field that current MLLMs still fall short on without a clear solution. This paper explores the usage of long-thinking to solve this problem and see some benefits, which is valuable to the research community.

The writing is very clear and easy to follow, the experiments are thorough and compelete, demonstrating the effectiveness of the proposed dataset.

**Reasons To Reject:**

Althought the overall experiemnt is clear, some details are missing. The one that I curious the most is the number of input visual tokens in Table1, especially on V* benchmark, a dataset containing high-resolution images and questions about visual details. I believe the 48.51 perforamnce on V* is obtained by controlling the number of visual tokens to a certain threshold (i.e. the high-res image has to be down sampled to fit the number of visual tokens since it used native resolution naViT). Will the LongPerceptualThoughts model still surpass the Qwen2.5-VL-7B-Instruct when we don't set a budget limit for visual tokens?

LongCoT can also be harmful for MLLMs since they tend to hallucinate more when its output is longer[1]. It would be great to discuss about this aspect since the dataset is endowing such ability to MLLMs.

[1] Huang, Q., Dong, X., Zhang, P., Wang, B., He, C., Wang, J., ... & Yu, N. (2024). Opera: Alleviating hallucination in multi-modal large language models via over-trust penalty and retrospection-allocation. In Proceedings of the IEEE/CVF Conference on Computer Vision and Pattern Recognition (pp. 13418-13427).

---

> ### Author Response · Authors · 2025-06-02
> **Response to Reviewer EHkH**
>
> We appreciate the reviewer’s positive feedback on the clarity of the writing, the completeness of the experiments, and the importance of the research question addressed.
>
> In the followings, we address the reviewer's concerns and questions
>
> 1. Varying thresholds of #pixels in Table 1, especially on V* benchmark
>
> As clarified in Section E.1, we set the maximum resolution to 512×512 during both training and inference throughout the paper.
>
> To address the reviewer’s question regarding the removal of the visual token budget, we conducted an ablation by varying the inference resolution while keeping the training setup fixed at a maximum of 512×512.
> Using the same VLMs from Table 1, we find:
>
> - At 512×512 (aligned with training), our fine-tuned VLM achieves the largest gains.
> - Increasing inference resolution introduces a distribution shift, as test inputs may exceed the training-time resolution. Nonetheless, scaling up to 2048×2048 still yields over a 2-point accuracy improvement relative to Qwen2.5-VL-7B-Instruct.
>
> These results indicate that LongPerceptualThoughts remains effective even without a strict visual token budget. We will clarify this in the final version.
>
>
> | Benchmark: V* Bench                          | Max #pixels = 512x512 | Max #pixels = 768x768 | Max #pixels = 1024x1024 | Max #pixels = 2048x2048 |
> |----------------------------------------------|----------|----------|----------|----------|
> | Qwen2.5-VL -- Zero-shot                      |   48.51  |   62.88  |   67.03  |   76.41  |
> | LongPerceptualThoughts - SFT-then-DPO (Ours) |   60.31  |   67.05  |   73.6   |   78.82  |
>
>
>
>
> 2. Hallucination benchmark
>
> We appreciate the reviewer’s concern regarding potential hallucinations introduced by equipping models with additional "thinking" steps. To evaluate this, we adopt the POPE benchmark [1]. POPE is consists of 9,000 examples and specifically targets on object hallucination. It has three splits: random, popular, and adversarial. We follow the exact setup in Table 1 and report both F1 and accuracy on the full benchmark.
>
> As shown in the table below, our method, LongPerceptualThoughts, achieves consistently better performance than the baseline, despite the added thinking tokens. We hypothesize that this improvement stems from the injection of structured cognitive behaviors (see Fig. 3a), which might offer some benefit in managing hallucination.
> While we observe improvements on POPE, we acknowledge that the relationship between extended reasoning and hallucination remains an open area of investigation.
>
> We agree this is an important direction for developing reasoning-capable VLMs and will include this analysis in the final version.
>
>
> |                                              | POPE - F1 | POPE - Acc |
> |----------------------------------------------|-----------|------------|
> | Qwen2.5-VL -- Zero-shot                      | 86.92     | 87.07      |
> | LongPerceptualThoughts - SFT-then-DPO (Ours) | 88.3      | 88.39      |
>
>
> 3. Additional related work
>
> We thank the reviewer for highlighting relevant prior work [2]. Similar to [3,4], which improve vision-centric tasks by helping VLMs "see" better, our approach is complementary: it encourages the model to "think" longer by exploring inter-search mechanisms (see L37–L45).
>
> We will clarify this distinction and add a discussion to Section 4.
>
>
> [1] Li et al., Evaluating Object Hallucination in Large Vision-Language Models
>
> [2] Zhang et al., MLLMs know where to look: Training-free perception of small
> details with multimodal LLMs.
>
> [3] Penghao Wu and Saining Xie. V*: Guided visual search as a core mechanism in multimodal llms.
>
> [4] Shao et al., Visual cot: Advancing multi-modal language models with a comprehensive dataset and benchmark for chain-of-thought reasoning

---

> > ### Comment · Reviewer_EHkH · 2025-06-09
> >
> > Thanks authors for the response. It's good to see that longperceptual thoughts can still benefit the model's performance under high resolution (although the improvement deminishes).
> >
> > And also good to see the performance on POPE, however, POPE's questions are mostly yes/no, so I guess the model still answer shortly on it, but my concern of hallucination is based on longer thoughts. So it would be great to design some experiments evaluating model's hallucination when it generate longer thoughts.
> >
> > Overall I think the idea and the research direction is promising and worth pursuing, so I keep my positive score. Please make sure to include the additional experiment and discussion to the next version of the paper, thanks!

---

> > > ### Author Response · Authors · 2025-06-10
> > > **Response to Reviewer EHkH**
> > >
> > > Thank you for your thoughtful feedback and continued support. We appreciate your suggestion regarding hallucination under longer thoughts and agree it’s an important direction. We are excited to include the hallucination experiments and the high-res analysis in the revised version as suggested. Thanks again for your constructive comments.

---

### Decision · Program_Chairs · 2025-07-08

**Decision:**

Accept

**Comment:**

The paper proposes LongPerceptualThoughts, a synthetic dataset of 30,000 long-form CoT traces designed to equip VLMs with System-2 reasoning skills like verification, subgoal setting, and backtracking. A three-stage automatic pipeline is introduced, which consists of: 1) multiple-choice questions (MCQ) generation, 2) simple CoT extraction from a base VLM, 3) CoT expansion with a stronger reasoning LLM. A Qwen2.5-VL-7B model is fine-tuned on this data (supervised fine-tuning (SFT), and then direct preference optimization (DPO)), which yields gains on five vision-centric benchmarks and MMLU-Pro (text-only benchmark).

This paper received scores of 7, 6, 4, 4 before the rebuttal. After the authors’ response, all reviewers maintained their original scores. There was also some discussion between reviewers and authors, and the AC gave additional time for reviewers to further discuss.

Among strengths, reviewers acknowledge that "this paper explores the usage of long-thinking to solve this problem and see some benefits", that this work has "successfully constructed a solid SFT dataset, where previous methods failed to correctly construct the reasoning SFT data for new domains", and consider the "analysis of where the performance improvement comes from (including reasoning patterns such as subgoal setting, verification, and backtracking)" to be insightful. Additional positive results on CV Bench, MMVP and DocVQA were also provided during rebuttal.

Some concerns include:
- Lacks comparisons to existing visual reasoning methods related to CoT: the method is only compared to one baseline (Visual-CoT, provided during rebuttal) on a single benchmark, which limits the strength of the evidence; this was pointed out by reviewer WtLj, but authors did not further replied with clarifications.
- The lack of ablation studies and output analysis makes it difficult to identify what drives the improvements. Without these, the contribution is unclear and offers limited value to the research community; this is a concern raised by WtLj and V9Qt.
- Most benchmarks remain multiple-choice or contain short answers; reviewers question whether long-form reasoning is truly exploited; however, authors show DocVQA results during rebuttal.

After weighting both strengths and remaining concerns, this AC leans towards accepting this paper. Yet, it is important to understand where methods success or fail, and authors are encouraged to provide additional empirical evidence.